# The History of Infectious Diseases and Medicine

**DOI:** 10.3390/pathogens11101147

**Published:** 2022-10-04

**Authors:** Tatsuo Sakai, Yuh Morimoto

**Affiliations:** 1Department of Physical Therapy, Faculty of Health Science, Juntendo University, Tokyo 113-8421, Japan; 2Department of Radiological Technology, Faculty of Health Science, Juntendo University, Tokyo 113-8421, Japan; 3Department of Medical History, Faculty of Medicine, Juntendo University, Tokyo 113-8421, Japan; 4Department of Anatomy and Life Structure, Graduate School of Medicine, Juntendo University, Tokyo 113-8421, Japan

**Keywords:** infectious disease, history of medicine, traditional Western medicine, modern medicine, exact medicine

## Abstract

From ancient times to the present, mankind has experienced many infectious diseases, which have mutually affected the development of society and medicine. In this paper, we review various historical and current infectious diseases in a five-period scheme of medical history newly proposed in this paper: (1) Classical Western medicine pioneered by Hippocrates and Galen without the concept of infectious diseases (ancient times to 15th century); (2) traditional Western medicine expanded by the publication of printed medical books and organized medical education (16th to 18th century); (3) early modern medicine transformed by scientific research, including the discovery of pathogenic bacteria (19th century); (4) late modern medicine, suppressing bacterial infectious diseases by antibiotics and elucidating DNA structure as a basis of genetics and molecular biology (20th century, prior to the 1980s); and (5) exact medicine saving human life by in vivo visualization and scientifically verified measures (after the 1990s). The historical perspectives that these five periods provide help us to appreciate ongoing medical issues, such as the present COVID-19 pandemic in particular, and remind us of the tremendous development that medicine and medical treatment have undergone over the years.

## 1. Introduction

From ancient times to the present day, the most important mission of medicine and medical care has been to diagnose and treat illnesses. It is well known that medicine before the 18th century was fundamentally different both in medical perception and medical scientific technology from what followed [1,2]. Until the 18th century, the diagnosis and treatment of illness was similar to that of ancient times, that is, the medicine of Hippocrates and Galen. After the 19th century, the power of science was harnessed and the efficacy of diagnosis and treatment gradually increased, such that in recent decades, it has grown exponentially.

After recent studies on the history of medical education [3,4], we newly distinguished the history of past and current medicine into the following five historical stages, namely, classic Western medicine (ancient times to the 15th century), traditional Western medicine (the 16th to 18th century), early modern medicine (the 19th century), late modern medicine (1900 to the 1980s), and exact medicine (the 1990s to the present day), incorporating the recent advances in the history of medicine. In the present article, we review how historical medicine progressed along with the development of scientific principles, as seen through the lens of infectious diseases in each of the historical stages (Table 1). We also focus on the medical textbooks and he medical education systems in those periods for a better understanding of the description of infectious diseases in those historical documents.

Since the beginning of the year 2020, the COVID-19 pandemic caused by SARS-CoV-2 has not only inflicted severe damage on the world economy but also caused indescribable suffering to people around the world. Although this experience has been a very disastrous one, it has also provided us with an opportunity to once again reaffirm the relevant roles of medicine and medical care for the world. At the same time, it is also relevant to recall the emergence of these great infectious diseases that have occurred one after another in history as part of the common experience of humankind.

## 2. Classic Western Medicine (Ancient Times to the 15th Century)

The Western medical tradition began in ancient Greek and Rome since 5th century BC, long before the discovery of infectious pathogens and without the concept of infectious diseases. The medical documents of Hippocrates and Galen were handed down and compiled into medical textbooks in the Middle Ages. From the records they contain of several devastating plagues, their causative pathogens have been investigated and interpreted by modern researchers from the symptoms and the modes of progression described therein.

### 2.1. Origin of Classic Western Medicine in Ancient Greek and Rome

Hippocrates was an excellent physician in Ancient Greece. Around 70 documents associated to him and his students were assembled to form the so-called Hippocratic Corpus (Latin: *Corpus Hippocraticum*) [5]. Among them, the *Epidemiai* (ἐπιδημίαι) volumes I-VII dealt with acute feverish illness of both infectious and non-infectious causes [6] (see Appendix A).

Galen of Pergamon was active during the Roman Empire. He performed anatomical dissections on various kinds of animals by himself and mastered the principles of prominent ancient medical works [7] (see Appendix A). He wrote numerous treatises on various aspects of medicine, including general medicine, natural sciences (physics), anatomy, physiology, dietetics, pathology, semeiotics, and pharmaceutics, and he has also documented the Antonine Plague he had witnessed [8] (see Appendix A). Once the Galenic treatises were translated into Syriac and Arabic, they were then transmitted to the Orient after the division of the Eastern and Western Roman Empire in 395.

### 2.2. Threatening Nature of the Plagues

The medicine in this period had no effective measures against infectious diseases. The historical documents recorded several devastating plagues, including the Plague of Athens in the 5th century BC (epidemic typhus by *Rickettsia prowazekii*?, smallpox?), the Antonine Plague in the 2nd century (smallpox?), the Plague of Justinian in the 6th century (bubonic plague?), and the Black Death in the 14th century (bubonic plague by *Yersinia pestis*) (see Appendix A).

### 2.3. Classic Medical Textbooks and Education Systems in the Middle Ages

In the Middle Ages, medical education started at the Schola Medica Salernitana in Salerno, southern Italy in the latter half of the 10th century. Arabic and Greek medical writings were translated into Latin, and universities were organized to teach medicine in Montpellier and Paris, France, and in Bologna and Padua, Italy in the latter half of the 12th century, followed by other universities, including Heidelberg in Germany and Leuven in Belgium [9]. In the medieval universities, medicine was taught by the Scholastic method, which consisted of exposition (*lectio*) of annotated authoritative texts and disputation (*disputatio*) of rational arguments to demonstrate conclusions [10]. The most popular teaching materials were represented by the *Articella*, an anthology of medical treatises compiled at Schola Medica Salernitana, and Avicenna’s *Liber Canonis*, which was the synthetic exposition of classical Greek medicine [9,11] (see Appendix A).

## 3. Traditional Western Medicine in the 16th to 18th Century

Western medicine experienced major reform in three aspects, namely, in the scientific research in anatomy, detailed medical observations, and systematized medical education, boosted by the popularization of printed medical books in the 16th century by printers such as Gutenberg in Germany and Plantin and Moretus in Belgium. Medical doctors observed and recorded various infectious diseases in published medical documents. Mass-produced printed medical books, as a form of information technology, accelerated the accumulation of medical knowledge and improved medical care by a virtual cycle of observation, documentation, and information exchange.

### 3.1. Reform of Medicine in Anatomy and the Impact of Fabrica

Medical study was further enhanced by Andreas Vesalius’s *De humani corporis fabrica* (On the structure of the human body, 1543) [12]. *Fabrica* was the first book on the human anatomy and its publication was revolutionary in that it introduced a major shift away from the hitherto assumed target of medical studies to better understand the classical medical texts to embrace the new scientific investigation of phenomena in nature, including the human body. The human anatomy also facilitated the improvement of surgical procedures by Ambroise Paré (see Appendix A).

### 3.2. Medical Observations on Infectious Diseases

Physicians in this period observed patients closely and made written reports on the appearance of several infectious diseases, including syphilis, malaria, and smallpox [13,14,15]. These infectious diseases have long had significant impact on human life and on culture, economy, and society. Variolation for smallpox immunization had long been practiced in China and the Orient, introduced to Europe by Lady Montagu, until Edward Jenner established the first modern vaccination against this infectious disease in the late 18th century [16] (see Appendix A).

### 3.3. Reform of the Education Systems and Observations

Medical education was innovated by a new style of teaching by oral lectures replacing the previously popular Scholastic method. In many universities, medicine was taught in four subjects, namely, *Theoria*, *Practica*, *Anatomia et Chirurgia*, and *Botanica et Pharmatica* (Box 1) [3]. Physicians became eager to observe and record accounts of diseases and patients. The results of clinical observations were published frequently as *Consilia* (Advice), *Consultationes* (Consultations), and *Observationes* (Observations), and new diseases were reported repeatedly [17].

Box 1The four major subjects taught in medical education during the period of traditional Western medicine.(A) *Theoria* (theory of medicine), the theoretical foundations of medicine in five subdivisions, including *Physiologia*, *Pathologia*, *Semeiotica*, *Hygiena*, and *Therapeutica*(B) *Practica* (practice of medicine), the individual illnesses(C) *Anatomia et Chirurgia* (anatomy and surgery), the structure of the human body as a basis of medicine, and its application to surgical operations(D) *Botanica et Pharmatica* (botanics and pharmaceutics), botanical remedies, utilizing botanical or herb gardens for education and research

In traditional Western medicine, *Practica* books dealt with the diagnosis, therapy, and prognosis of individual illnesses, including both the regional illnesses *a capite ad calcem* (from head to toe) and systemic illnesses such as fevers [18,19]. Those illnesses corresponding to today’s infectious diseases were allocated to various sections of the *Practica* books, indicating there was no collective concept of infectious disease at that time (see Appendix A).

## 4. Early Modern Medicine (19th Century)

Western medicine was transformed into modern medicine from the 19th century on by scientific research in the basic medicine and technical innovations in the clinical medicine. To outline the transformation of medicine, we will focus on pathological diagnosis, surgical innovation, and the discovery of pathogens.

### 4.1. Discovery of Physical Diseases by Pathological Anatomy

Advances in medicine in the 19th century led to a paradigm shift that changed the concept of disease and improved medical treatment. First, the advent of pathological anatomy changed the concept of disease. Before the 18th century, without pathological anatomy, the symptoms, including systemic symptoms (fever), gastrointestinal symptoms (diarrhea and vomiting), respiratory symptoms (cough and dyspnea), and so on, were confused with the diseases and represented the illness. In the 19th century, the pathological changes of organs found by postmortem autopsy were regarded to be the cause of diseases, or the illness itself. There are several diagnostic innovations observed during this time [20] (Box 2).

Box 2Diagnostic innovations in early modern medicine (19th century).Jean-Nicolas Corvisart: Discovery of the diseases of the heart and great arteries conducted in vivo diagnosis using the chest percussion maneuver (1806).René Théophile Hyacinthe Laënnec: Development of indirect auscultation and discovered pulmonary diseases supported by observations made during autopsy (1819).Richard Bright: Discovery of the kidney diseases by autopsy of dropsy patients after scarlet fever (1827), which came to be known as Bright’s disease.Rudolf Ludwig Carl Virchow: Cellular pathology (1858) and recommended histopathological diagnosis, which improved the accuracy of pathological diagnosis.

### 4.2. Sterilization Contributed to the Innovations in Surgery

Surgery up to the 18th century had high risks of developing bacterial infections peri- and post-operationally, and its application was limited to the treatment of wounds and tumors on the body surface. The British surgical scientist Joseph Lister proposed a disinfection method using carbolic acid in the 1870s and later developed a surgical sterilization method. In the mid-19th century, ether anesthesia was performed in a public surgery in the United States (1846), and the anesthesia method spread rapidly to Europe and the United States. These innovations enabled surgical operations of visceral organs deep in the body, such as the digestive, respiratory, and reproductive organs (Box 3).

Box 3Key persons and the innovations in surgery during the period of early modern medicine.Joseph Lister: A disinfection method using carbolic acid
in the 1870s and later developed a surgical sterilization method.Theodor
Billroth: Pyloric resection (1881) and gastrectomy (1885)William Halsted:
Development of radical mastectomy for breast cancer (1889)Harvey Cushing: A
pioneer of brain surgery (from 1912)

### 4.3. Discovering Pathogens of Infectious Diseases

With the Industrial Revolution and urbanization in the 19th century, the water-borne diseases of dysentery and cholera became common causes of death in many countries [21] (see Appendix A).

In the mid-19th century, the causes of plagues and epidemics were still obscure, and miasma (contaminated air) and contagion (disease seeds) were discussed as possible causes of diseases. The German anatomist Jakob Henle (1840) classified the causes of diseases into miasmas, contagions, and miasmatic-contagions [22]. The concepts of “miasma” from the ancient Greek “μίασμα” (pollution) and “contagion” from the Latin “*contagio*” (contact) had been used since the 16th century [23].

In the latter 19th century, the French chemist and microbiologist Louis Pasteur disproved the spontaneous generation theory (1861) [24] and revealed that invisible microorganisms were the cause of spoilage and fermentation to support the germ theory of disease (1878) [25]. The German physician and microbiologist Robert Koch discovered *Bacillus anthracis* [26], *Mycobacterium tuberculosis* (*M. tuberculosis*) (1882) [27], and *Vibrio cholerae* (*V. cholerae*) (1884) [28], all three of which were important pathogens of major infectious diseases. Alexandre Yersin and Shibasaburo Kitasato discovered *Yersinia pestis* (*Y. pestis*), the pathogen of the plague (1894). Bacteriology was the mainstream state-of-the-art science, leading to the discovery of numerous pathogens during this period (Table 2).

### 4.4. Infectious Diseases in Medical Textbooks

In the 19th century Europe, medical textbooks dealing with the diagnosis, therapy, and prognosis of individual diseases were published under various titles such as “practice of medicine” in English, “*specielle Pathologie*” in German, and “*pathologie interne*” in French. In these textbooks, the diseases of organs were arranged according to individual organ systems in addition to systemic diseases. As the most popular textbook in this period, Niemeyer’s *Lehrbuch der speciellen Pathologie und Therapie* (1858) [29] recognized various acute and chronic infectious diseases in the section of “constitutional diseases” (volume 2, part 2, Section 3) (see Appendix A).

## 5. Late Modern Medicine (20th Century)

In the 20th century, the increased international mobility and the worldwide wars spread serious airborne infectious diseases, such as influenza and tuberculosis. On the other hand, medical science succeeded in overcoming the infectious diseases by antibiotics against bacterial pathogens and by identifying specific pathogenic viruses.

### 5.1. Antibiotic Development

The investigation of antimicrobial agents began with the discovery of pathogenic bacteria at the end of the 19th century. Salvarsan and sulfonamides were produced by chemical synthesis in 1911 and 1935, respectively [30]. Alexander Fleming discovered penicillin in 1928, and the compound was mass-produced as its initial molecule, benzylpenicillin [31]. The discovery of penicillin opened the door to the golden age of exploration of new antibiotics from natural compounds, and natural compounds and semi-synthetic derivatives of those compounds were commercialized and introduced into clinical settings [32]. Many bacterial infections that have plagued humankind since ancient times have been treated by those “silver bullets” (Figure 1 and Figure 2).

Bacterial infectious diseases were significantly suppressed by antibiotics, and deaths from infectious diseases such as tuberculosis and gastroenteritis decreased sharply after World War II [33]. Penicillin-resistant *Staphylococcus aureus* (*S. aureus*) was, however, reported in 1942, the same year of the introduction of Penicillin G [34]. This led to the development of methicillin, an antibiotic that is not hydrolyzed by penicillinase. However, methicillin-resistant *S. aureus* (MRSA) was first reported in 1961 [35], and rapidly spread worldwide in the 1970s [36] (Figure 3).

### 5.2. Discovery of Viruses

At the end of the 19th century, while bacteria were discovered, viruses such as influenza and smallpox were not yet known. Tobacco mosaic virus was the first virus to be discovered in 1892 by Dmitri Ivanovsky [35]. Dutch microbiologist Martinus Beijerinck observed that the reproduction of infectious matter was associated with a living host cell and named the agent *contagium vivum fluidum* (living infectious fluid) [36]. At the time, a virus was considered a filterable entity, and several major viral pathogens were isolated [38].

The electron microscope, invented in 1938 by German scientists Max Knoll and Ernst Ruska, enabled the observation of virus morphology [39]. During the 1950s, increasing knowledge about DNA and heredity after the double helix model by Watson and Crick [40] revealed the functional mechanism of the virus [38] (Table 3).

### 5.3. The Major Airborne Diseases

By the mid-19th century, the records of six influenza pandemics remained (1510?, 1557?, 1580, 1729, 1781-82, and 1830-33) [41,42], and detailed records have been kept since the late 19th century with increased awareness of public health. Tuberculosis has been known to be around since prehistoric times by the tracing of lesions left in mummies [41]. By the end of the 19th century, the disease reached epidemic proportions in Europe and North America [43]. The introduction of streptomycin and BCG vaccination programs dramatically reduced the number of patients [44] (see Appendix A).

### 5.4. Infectious Diseases in Medical Textbooks and Academic Advances in Molecular Biology

In the early 20th century, infectious diseases increasingly attracted the attention of researchers after the discovery of pathogenic bacteria at the end of the 19th century. The contents of medical textbooks in this period generally listed the infectious diseases at the top, followed by diseases of organs in various organ systems, as exemplified by the most notable textbook of the time, Osler’s *Principles and Practice of Medicine* (1st ed. 1892, 9th edition, 1920) [45] (see Appendix A).

## 6. Exact Medicine (1990–to Date)

Since the 1990s, modern medicine has witnessed a giant leap forward to establish “exact medicine”. Previously, clinical treatments had been based on empirical knowledge, and the final diagnosis was made by postmortem autopsy. Patients accepted the treatment decision made by physicians with implicit trust. Exact medicine from the 1990s achieved accurate diagnosis by medical imaging, evidence-based medicine, and a treatment decision based on informed consent. The four characteristics of exact medicine are described in Box 4.

Box 4The four characteristics of exact medicine.
In vivo visualization: The diseases are diagnosed in vivo by medical imaging, instead of postmortem autopsy.Scientific verification: The therapeutic measures are evaluated by clinical research and standardized publicly, instead of by personal empirical evaluation.Partnership with patients: Patient-centered care is based on an equal relationship between physicians and patients with explicit explanation, instead of implicit mutual trust.Cooperation in clinic: Medical practice is performed by a team of medical staff members from multiple disciplines, instead of by one physician alone.


### 6.1. Antiviral Drugs

The first antiviral agent with a specific target was aciclovir (ACV or acyclovir), initially introduced in 1977 for the treatment of herpes simplex virus infections caused by HSV-1 (herpes simplex virus type 1) [46,47]. Soon after the isolation of the human immunodeficiency viruses (HIV) in 1983 by Françoise Barré Sinoussi, the first antiretroviral agent, azidothymidine (AZT), was developed, followed by the rapid development of other antiretrovirals [48]. The number of targeted infections also increased to include influenza virus infections, herpesvirus infections, human cytomegalovirus infections, hepatitis virus infections (hepatitis B and C), varicella-zoster virus infections, HIV infections, and respiratory syncytial virus infections [48].

### 6.2. Infectious Diseases in the Period of Exact Medicine

Seen from the perspective of the history of medical science, current medicine obtained robustness in the diagnosis and treatment of infectious diseases. Even though human society faces emerging issues such as the ongoing pandemic, the global burden of antimicrobial resistance remains one of the most urgent health threats. AIDS is still the leading cause of death across Sub-Saharan Africa, although antiviral drugs have been developed. Since the beginning of the 21st century, three coronavirus pandemics (SARS-CoV, MERS-CoV, SARS-CoV-2) have emerged, the present one of which is still ongoing [49] (see Appendix A).

### 6.3. One Health Concept

The outbreaks occurring prominently during the period of exact medicine are associated with interactions between humans and domestic and wild animals. In fact, approximately 75% of emerging human diseases in the period of late modern medicine and exact medicine have been zoonotic [50,51]. One Health is a holistic approach that looks at health in the context of human, animal, and environment relationships [52], developed from the concept of One Medicine proposed by Calvin Schwabe [53], and formulated in 2004 [54]. One Health has been developed and implemented in the major international organizations in the last two decades [52], and involved multiple disciplines, including public health, veterinary medicine, and environmental sectors. In 2021, the One Health High-Level Expert Panel (OHHLEP), which addresses health crises arising from the human–animal–ecosystem interface, was launched [55].

### 6.4. Infectious Diseases in Medical Textbooks (1990s–to Present)

Since the end of the 20th century, various novel categories of diseases were recognized in addition to infectious diseases and diseases of the organs, including malignant tumors, autoimmune diseases, endocrine and metabolic disorders, and diseases caused by environmental factors. These topics were included in the medical textbooks in this period as exemplified by the widely adopted, best-selling American textbook *Harrison’s Principles of Internal Medicine*, 20th edition (2018) [56] (see Appendix A).

## 7. Conclusions

By following the history of medicine since the ancient Greek and Roman period and the severe experiences with various infectious diseases, we can better appreciate the current state of medical scientific achievement. When seen through the historical viewpoint on infectious diseases, advanced “exact medicine” has greatly minimized the impact of COVID-19, which has caused significant damage to society and humanity. The fear of COVID-19 has been remarkably reduced thanks to exact medicine, and human civilization must continue to co-exist with SARS-CoV-2. The world in the post-COVID era may never be the same as before, as the historical infectious diseases changed society, economy, and human life in each epoch. It is evident that novel world health issues will continue to emerge. In 2022, as the COVID-19 pandemic continues, monkeypox outbreaks have been reported from non-endemic and endemic countries in widely disparate geographical areas [57]. Further historical studies from the medieval quarantine to the modern antibiotic and antivirals, especially on vaccine discovery, diagnostic methods, and the challenges in the medical field, are warranted to evaluate the era of “exact medicine”. The historical perspective of medicine in the five time periods proposed in this review guide us in better appraising the significance, expectations, and limitations of ongoing medical issues.

## Figures and Tables

**Figure 1 pathogens-11-01147-f001:**
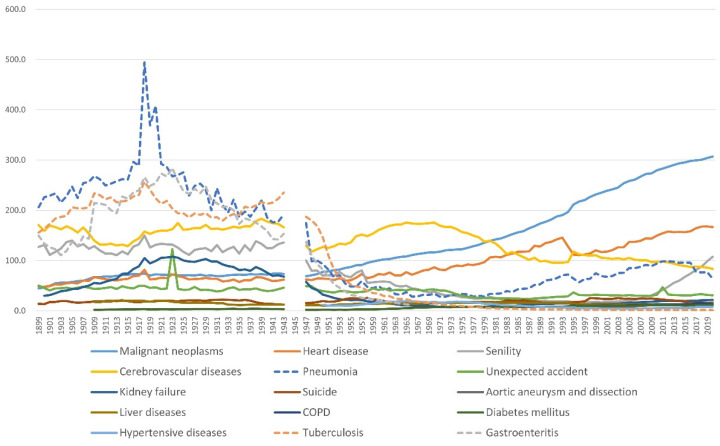
Annual transition of the mortality rate by cause of death in Japan from 1899 to 2020. The infectious diseases (dotted lines; pneumonia, tuberculosis, and gastroenteritis) were the top three causes of death before 1945 and decreased rapidly after 1945. Mortality per 100,000. From the data of vital statistics, Ministry of Health, Labour, and Welfare, prepared by T.S.

**Figure 2 pathogens-11-01147-f002:**
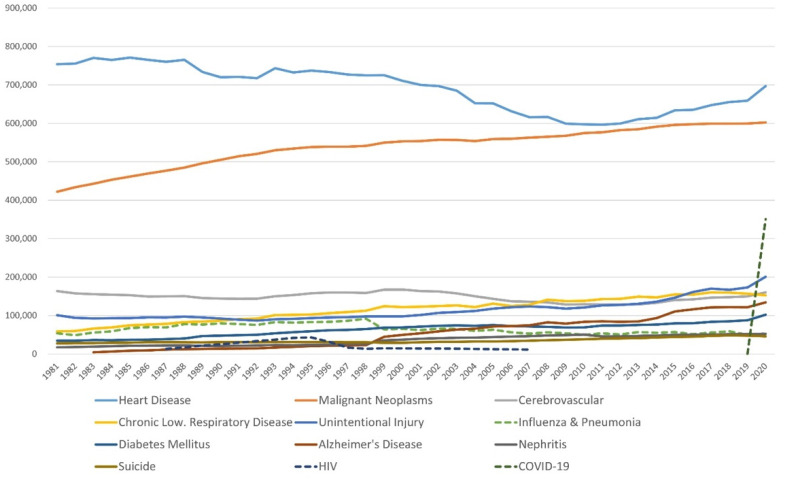
Annual transition of the number of deaths by cause of death in the USA from 1881 to 2020. Influenza and pneumonia (green dotted line) was in 6th place in the ranking of average mortality in this period. HIV (dark blue dotted line) was in 8th place from 1992 to 1996. COVID-19 (dark green dotted line) was in 3rd place in 2020. From the data of WISQARS^TM^, Centers for Disease Control and Prevention, prepared by T. S.

**Figure 3 pathogens-11-01147-f003:**
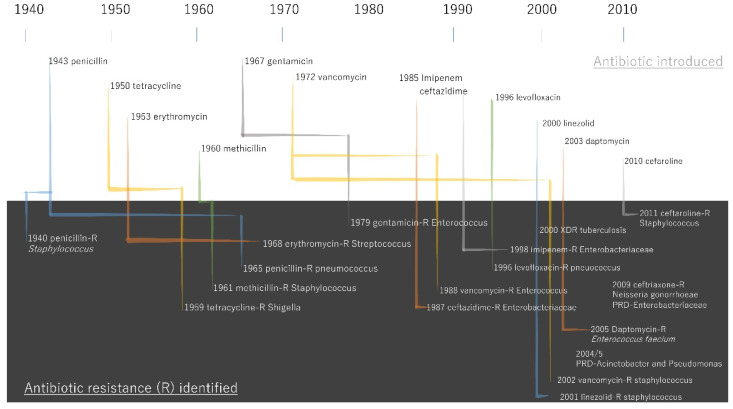
Timeline of antibiotic introductions and key antibiotic resistance events. The data was obtained by the report “Antibiotic Resistance Threats in the United States, 2013.” [37].

**Table 1 pathogens-11-01147-t001:** Five periods in the history of medicine.

History of Medicine and Medical Education	Infectious Diseases	Diagnosis and Treatment
**Western Classic Medicine** (Ancient–15th century)	**Plague of Athens** (BC429–) **Antonine Plague** (165–) **Plague of Justinian** (541–) **Black Death** (1347–)	**Diagnosis and prognosis by signs** (Uroscopy, Pulse-reading) **Hygienic therapy** (Regimen, Herbals, Venesection)
**Ancient medical documents** (Hippocrates, Galen) **Scholastic education** (*lectio*, *disputatio*)		
**Traditional Western medicine** (16th–18th century)	**Syphilis** (1494–) **Malaria** **Smallpox** (Vaccination, 1798)	
**Anatomical research** **Lecture-based education** (*Theoria*, *Practica*, Anatomy/Surgery, Botany/Pharmaceutics) **Clinical observations**		
**Early Modern Medicine** (19th century)	**Cholera** (Pandemic, 1817–) **Dysentery**	**Diagnosis by autopsy after death** **Innovation of surgery** (Anesthesia, Disinfection) **Discovery of pathogenic bacteria**
**Basic medicine** (Anatomy, Physiology, Pathology, Pharmacology, Hygiene) **Clinical medicine** (Internal medicine, Surgery, etc.) **Cell theory** (1838–1839)		
**Late Modern Medicine** (1900–1980s)	**Influenza** (Spanish flu, 1918–1920) **Tuberculosis**	**Antibiotics** **Discovery of viruses**
**Basic research** (Bacteriology, Biochemistry, Cell biology) **Antibiotics** (penicillin, 1941)		
**Exact Medicine** (1990s–)	**Antimicrobial resistance** **AIDS** (1981–) **COVID-19** (2019–)	**Diagnosis by medical imaging in vivo** **Scientific verification** **Partnership with patients** **Clinical cooperation**
**Basic research** (Molecular biology, Immunology) **Clinical and translational research**		

**Table 2 pathogens-11-01147-t002:** Discovery of pathogenic bacteria in early modern medicine and the beginning of late modern medicine.

Disease	Pathogen	Year	Discoverer
Leprosy	*Mycobacterium leprae*	1874	Hansen, Gerhard Armauer (Norway)
Anthrax	*Bacillus anthracis*	1876	Koch, Robert (Germany)
Typhoid fever	*Salmonella enterica*	1880	Eberth, Karl Joseph (Germany)
Tuberculosis	*Mycobacterium tuberculosis*	1882	Koch, Robert (Germany)
Cholera	*Vibrio cholerae*	1883	Koch, Robert (Germany)
Diphtheria	*Corynebacterium diphtheriae*	1883	Klebs, Edwin (Switzerland)
Tetanus	*Clostridium tetani*	1884	Nicolaier, Arthur (Germany)
Brucellosis	*Brucella* sp.	1887	Bruce, David (Great Britain)
Plague	*Yersinia pestis*	1894	Yersin, Alexandre (France); Kitasato, Shibasaburo (Japan)
Dysentery	*Shigella dysenteriae*	1897	Shiga, Kiyoshi (Japan)
Syphilis	*Treponema pallidum*	1905	Schaudinn, Fritz; Hoffmann, Erich (Germany)
Whooping cough	*Bordetella pertussis*	1906	Bordet, Jules (France)
Epidemic typhus	*Rickettsia prowazekii*	1909	Nicolle, Charles (France)

**Table 3 pathogens-11-01147-t003:** The representative pathogenic viruses discovered in late modern medicine.

Disease	Pathogen	Year	Discoverer
Polio (Poliomyelitis)	Poliovirus	1909	Landsteiner, Karl (Austria)
Yellow fever	Yellow fever virus	1928	Stokes, A; Bauer, JH; Hudson, NP (USA)
Influenza	Influenza viruses	1933	Smith, W; Andrews CH; Laidlaw PP (USA)
Japanese encephalitis	Japanese encephalitis viruses	1935	Hayashi, Michitomo (Japan)
Shingles (Herpes zoster)	Varicella zoster virus (VZV)	1953	Weller, Thomas Huckle (USA)
Measles	Measles virus	1954	Edmonston, David (USA)
Rubella	Rubella virus	1962	Parkman, Paul Douglas (USA) Weller, Thomas Huckle (USA)
Hepatitis B	Hepatitis B virus	1966	Blumberg, Baruch Samuel (USA)
gastroenteritis	Noro virus	1972	Kapikian, Albert Z. (USA)
Acquired immunodeficiency syndrome (HIV/AIDS)	Human immunodeficiency virus (HIV)	1983	Gallo, Robert (USA); Montagnier, Luc (France)
Hepatitis C	Hepatitis C virus	1989	Choo QL, Kuo G, Weiner AJ, Overby LR, Bradley DW, Houghton M (USA); Kuo G, Choo QL, Alter HJ, Gitnick GL, Redeker AG, Purcell RH, Miyamura T, Dienstag JL, Alter MJ, Stevens CE et al. (USA)

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
