# Peer review of "The History of Infectious Diseases and Medicine"

_pathogens, 2022, doi:10.3390/pathogens11101147_

Round 1
Reviewer 1 Report
In this article, Sakai and Morimoto analyzed and reviewed various historical and current infectious diseases in a five-period scheme of medical history they established: 1) Classical Western medicine (Hippocrates and Galen), 2) Traditional Western medicine, 3) Early modern medicine, 4) Late modern medicine and 5) Exact medicine. They stated the historical perspectives that these five periods provide can help us to appreciate ongoing medical issues and remind us the huge development of medicine and medical treatments. I can agree with that statement. It is indeed useful and inspiring. Overall, this unusual type of review is interesting and quite well-written. Please find below a detailed list of my moderate remarks/suggestions.
Strengths: Interesting and useful subject (especially for teachers), original, and quite well-written.
Weakness: Some famous scientists are missing and small elements could be added in some sections. Several references should be added for some sentences.
Major
/
Moderate
-L13: The originality of the five-period scheme should be further emphasized.
-L18: The elucidation of DNA structure and the development of genetics and microbial genetics should be mentioned.
-L46: It should be mentioned that COVID-19 is caused by SARS-CoV-2.
-L48: “most” should not be used here. The impact of the black plague (especially year 1348) was far more significant. The flu in 1918 has a very significant impact too.
-L54: Please add dates.
-L64: Please add a reference.
-L79-81: The cause when identified should be mentioned (plague is a vernacular word). Black plague (1348…) was caused by Yersinia pestis.
-2.3: Leuven (Louvain) established in the 15th century should be mentioned too as it was one of the place of Vesalius (with Montpellier).
-L99: Gutenberg and Moretus Plantin, two famous printers should be mentioned.
-L115-116, notably in China. Please add it. Please also do not forget Lady Montagu who had a significant impact too, before Jenner.
-In the 16th century a surgeon as Ambroise Paré had an important impact too with the development of ligatures to treat injuries instead of cauterization.
-L166-173: Alexandre Yersin and Dr Kitasato should be mentioned here too for Yersinia pestis.
-L191: Please add a reference.
-L211: Please add a reference. L213 too.
-L213: The names Crick and Watson should be mentioned.
-L242: Please add the HSV-1 (herpes simplex virus type 1) acronym and define it.
-L243: Françoise Barré Sinoussi (still alive) should be mentioned too.
-L245: Please add a reference.
-L245-247: Please do not mix virus names and disease names.
-L242: Please add a reference.
-L254: Please add a reference.
-L255: Please provide more details about these three coronavirus pandemics (more limited fortunately)
-L256: Please add a reference.
6.2-6.3: The One Health concept must be presented too. A more holistic approach could make the difference. Many pathogens have animal origins, see Lancet 2012; 380: 1956–65 and https://doi.org/10.1016/j.animal.2021.100241 for instance.
-L261: Please add a reference.
-In Fig1: What are the units for the y axis? The legend is too small. The figure could be improved.
Minor
-L19: Life and health, is not a little bit redundant?
-L19: In vivo should be in italic.
-L82: The dot should be at the end of the sentence, after the brackets. See also L95, L110, L117, L131…
-L272: I would rather mention the virus (SARS-CoV-2) here and not the disease. “Monkey” Poxvirus could be mentioned too.
-L280: There is something wrong?
-In table 1 and in the text: All the Latin words should be in italic.
-P8: In vivo should be in italic.
-Daniel Elmer Salmon could be mentioned too for Salmonella.
-Bowes 1, 2 and 4: The dots should be removed at the end of the “sentences” as they are not actually sentences (no verbs).
-References: 1 and 2, pages are missing. References 7 and 8 and others, the style for journal names is not always the same, please harmonize. See also 29, 30…
-L32: Please correct Staphylococcus.
Author Response
Please see the attachment. The PDF file contains the response to reviewer 1 followed by the main text.

Reviewer 2 Report
Review article by Tatsuo Sakai et al. discussed and distinguished into five historical stages of the history of Infectious disease starting from ancient times through medieval ages through modern medicine. And also, how historical perspectives of infectious disease impacted global economy and development of medicine. Finally concluded with COVID-19 impact and how historical perspective of medicine guide us in appraising the significance,expectation and limitations of ongoing medical issues.
The manuscript was well written; however, this review did not give any interesting future directions apart from concluding with already established facts in the field. It would be appropriate for the review article to add appropriate conclusion to the history of infectious disease and medicine and future direction if necessary.
Below are suggestions for minor revision of the manuscript:
1. This review article title was not justified thoroughly by confining the conclusions to COVID-19. Authors would have discussed global impact of infectious diseases and future directions rather than ending the conclusion with COVID-19.
2. It would have been appropriate to include global burden of infectious diseases by adding the WHO and CDC data on infectious disease and mortality rate along with Fig -1.
3. Adding timeline of antimicrobial drugs (antibiotics and etc) and drug resistance mechanisms would have been great addition.
4. Starting from quarantine to antibiotic, vaccine discovery diagnostic methods and challenges in the medicine field throughout the history of medicine needs to be discussed more elaborately.
I have no major concerns with this paper
Author Response
Please see the attachment. The PDF file contains the response to reviewer 2 followed by the main text.

Round 2
Reviewer 1 Report
The article has been improved. However, some small modifications are still need.
-L18: A reference should be added for DNA structure elucidation.
-L46: Sorry for the small mistake, please write SARS-CoV-2 instead of SARSCoV-2.
-L48: English could be improved.
-L79-81: Typhus is not a cause but the name of a disease (not even a scientific name), please give the name of the pathogen.
-Leuven is in Belgium now, please correct.
-Moretus PLantin printer is in Belgium too, please correct.
-6.3: not only during the period of exact medicine, before also...
